# The Power of Recognition: A Qualitative Study of Social Connectedness and Wellbeing through LGBT Sporting, Creative and Social Groups in Ireland

**DOI:** 10.3390/ijerph16193636

**Published:** 2019-09-27

**Authors:** Nerilee Ceatha, Paula Mayock, Jim Campbell, Chris Noone, Kath Browne

**Affiliations:** 1School of Social Policy, Social Work and Social Justice, University College Dublin, Belfield, Dublin 4, Ireland; jim.campbell@ucd.ie; 2School of Social Work and Social Policy, Trinity College, Dublin 2, Ireland; pmayock@tcd.ie; 3School of Psychology, National University of Ireland Galway, Galway, Ireland; chris.noone@nuigalway.ie; 4School of Geography, University College Dublin, Belfield, Dublin 4, Ireland; kath.browne@ucd.ie

**Keywords:** LGBT, wellbeing, Ottawa Charter, recognition, Theory of Recognition, mental health, social inclusion and sense of community, social participation, community participation, social connectedness, community connectedness

## Abstract

The broad research consensus suggesting substantial vulnerabilities among lesbian, gay, bisexual and transgender (LGBT) communities may fail to recognize the protective factors available to these populations. The sparse literature on mental health promotion highlights the importance of understanding strengths-based community approaches that promote LGBT wellbeing. Informed by the *Ottawa Charter for Health Promotion*, underpinned by Honneth’s *Theory of Recognition*, this paper outlines the findings of a qualitative Irish study on LGBT social connectedness through a diverse range of sporting, creative and social interests. Ten in-depth interviews were conducted with 11 people (including one couple) who self-identified as lesbian (5), gay (4), bisexual (1) and transgender (1) aged between 22 and 56 years. A university Research Ethics Committee granted approval. The data were transcribed and coded using thematic analysis, enhanced through a memo-writing approach to reflexivity. The theme of ‘connecting’ emphasized the shared nature of activities, with like-minded others through groups established by, and for, LGBT communities. Messages from the study reinforce the central role of LGBT communities in the promotion of mental health and social wellbeing, with important policy and practice implications. This requires the contextualization of the contribution of LGBT communities within understandings of social justice, identity and recognition.

## 1. Introduction

According to the World Health Organization (WHO) “wellbeing is fundamental to quality of life, enabling people to experience life as meaningful and to be creative and active citizens” [1]. The WHO makes a further contribution to our understanding of wellbeing through the *Ottawa Charter for Health Promotion,* which defines health promotion as: “the process of enabling people to increase control over, and to improve, their health” [2] The *Charter* emphasizes the need to both change the environment and strengthen the person, acknowledging that:
*People cannot achieve their fullest health potential unless they are able to take control of those things which determine their health*.[2]


Social connectedness has been identified as a pivotal concept in approaches to mental health promotion and core to wellbeing, as articulated in *Connecting for life: Ireland’s national strategy to reduce suicide 2015–2020* [3]. This strategy seeks to promote a society “where communities and individuals are empowered to improve their mental health and wellbeing” [3]. It highlights protective factors, such as social connectedness, in reducing isolation and promoting help-seeking. The Strategy’s second goal is to strengthen community capacity, in recognition that:
*An empowered community can respond to the needs of its members and protect them in difficult times and can sustain these positive effects over time*.[3]


While *Connecting for Life* adopts a whole population approach, it also identifies specific priority groups, notably those vulnerable to suicide, including lesbian, gay, bisexual and transgender (LGBT) communities. Such policies tend to refer to a broad, albeit contested, notion of heightened LGBT mental health risk, typically contextualized within a minority stress framework [4], describing the consequences of discrimination against, and victimization of, minority groups [3,4,5]. Thus, it is contended that:
*LGBT people are at a heightened risk of psychological distress because of the stresses created by stigmatisation, marginalisation and discrimination*.[6]


In this way, *Connecting for Life* aligns with the five-pronged approach outlined in the *Ottawa Charter* [2,6]. As such, the potential of the interconnection of healthy policies and supportive environments, which enable strong connected communities and support individuals’ agency in relation to their wellbeing [2], is of particular relevance for LGBT populations [3,4,5,6].

The concept of social justice has been identified within the *Ottawa Charter* as one such prerequisite for health [2]. While there are multiple definitions of the concept, distinctions have been drawn between redistributive and recognitive forms of justice. Redistributive justice identifies the political-economic structures as causing social injustice with Fraser expanding the concept to include recognition: “*Cultural injustice is rooted in social patterns of representation, interpretation and communication* [7]. Fraser’s notion of “recognitive justice” [7] highlights the importance of revaluing disrespected identities and argues for recognition of the cultural axis of injustice in order to promote cultural diversity through group differentiation [7]. Honneth, drawing on the work of Hegel, Mead and Taylor, further developed these ideas on recognition:
*The moral quality of social relations cannot be measured only in terms of the fair and just distribution of material goods, rather, our notion of justice is also linked very closely to how, and as what, subjects mutually recognise each other*.(p. 17). [8]


In Honneth’s *Theory of Recognition,* such forms of recognition are critical in revaluing disrespected identities [9]. Honneth applies his framework of social justice to emancipation struggles which can promote cultural diversity through group differentiation [10]. In this way, the struggle for recognition provides the impetus for social change [8,9]. Importantly, claims for rightful identity recognition are the motivation for social transformation [9]. This application includes activism by LGBT communities described as:
*culturally integrated communities with a common history, language and sensibility…[who] developed a self-understanding…a transformation of collective self-understanding…that could lead to the claim for recognition of one’s own culture…The concept of “identity politics” captures this idea*.[10]


The tripartite framework developed by Honneth has three interlinking spheres of recognition: intersubjective recognition; recognition of individual members through community and social networks; and legal recognition of universal rights [9]. This tripartite framework is outlined in Table 1 below:

Honneth argued that intersubjective recognition is reciprocal in that there is mutual recognition by others whom one also recognizes. This supports the development of security and resilience: “The certainty about the value of one’s own needs can be called “self-confidence” [9]. However, the *Theory of Recognition* both encompasses and surpasses interpersonal relationships to include the recognition of the unique contribution of community members and legal recognition of universal human rights [9]. He identified that recognition is through community relations where: “the individual is recognized as a person whose capabilities are of constitutive value to a concrete community” [9]. This acknowledgment results in self-worth. Honneth identified that recognition necessarily required forms of legal relations involving rights that have “the character of universal human treatment”. This, in turn, promotes empowerment whereby “such a type of certainty about the value of one’s own judgment can be called self-respect” [9].

In this way, Honneth emphasized the importance of recognition for LGBT people and communities, highlighting the potential of the social environment to enable strong, connected communities that support LGBT wellbeing [2]. This provides a basis for evaluating whether LGBT communities are supported through interpersonal, community and legal recognition in order to gain control over health, which is fundamental to health promotion [2,9]. This is of particular relevance given that policy approaches frequently recognize that LGBT people are at heightened mental health risk [3,4,5]. On the other hand, a number of researchers have cautioned against such overly negative discourses which tend to offer limited framing of LGBT lived experience, particularly that of LGBT youth [11,12,13,14,15]. It is important, therefore, to acknowledge alternative narratives that explore positive connotations of mental health and social wellbeing, consistent with the *Social Determinants of Health*:
*As social beings, we need…to feel valued and appreciated…Belonging to a social network of communication and mutual obligation…has a powerful protective effect on health*.[16]


The predominant focus on LGBT mental health risk has tended to obscure alternative approaches that emphasize LGBT strengths-based community approaches to LGBT social connection and wellbeing [17]. It has been argued that “current conceptualizations fail to explain why many LGBT people enjoy good health despite adversity” (p. 5). [17]. Hass et al., while confirming the consensus of health inequalities for LGBT communities, equally acknowledge the limitations of the existing research base, arguing that “(r)elatively little research has been done on factors that protect the large majority of LGB people from suicidal behavior” [3] (pp. 26). They go on to recommend that future research priorities should:
*Conduct studies of factors that protect against or mitigate the impact of suicide risk factors…and factors that contribute to the development of resiliency…studies should also include potentially protective factors such as…community connectedness*.[3]


In conclusion, the important principles of the *Ottawa Charter*, underpinned by Honneth’s *Theory of Recognition* [8,9,10], offers the the potential for novel approaches to social and community connectedness that highlight more positive aspects of LGBT individual and collective wellbeing. This will now be explored in relation to the following: healthy policies; supportive environments; community action; personal skills; and reorienting health services [2].

### 1.1. Building Health-Promoting Policy

LGBT communities have been at the forefront of long-standing activism in relation to legal recognition in Ireland [18]. While homosexuality was not decriminalized until 1993, there has since been greater recognition of LGBT rights. In 2014, at the *International Day Against Homophobia and Transphobia*, the Irish Government became a signatory to the non-binding *Declaration of Intent,* which asserts that:
*Human beings are entitled to the full enjoyment of all human rights, regardless of sexual orientation and/or gender identity*.(p. 1) [19]


This commitment informed legislative and policy measures including those seeking to combat discrimination. In 2015, a referendum was passed which amended the Irish Constitution, providing for marriage-equality legislation [20]. That same year, Ireland enacted the Gender Recognition Act [21], with a review making further recommendations for people under 18 years of age [22]. Such initiatives may provide a foundation for a society committed to respecting diversity and empowering LGBT communities, as advocated by Honneth [8,9,10]. However, it is not clear how well such ideas have been translated into health policies and the promotion of wellbeing, and specifically, in relation to protective factors for LGBT communities. *Connecting for Life* asserted that:
*…research [is] under-developed in the Irish context, and identified as [a] priority: focus on the protective factors for mental health…and how these can be ameliorated within prevention programmes*.[3]


Further, the publication of the *LGBTI+ National Youth Strategy* as part of the Irish programme government acknowledged the “limited availability of Irish-specific data, statistics and…research relating to young LGBTI+ people in Ireland and, more broadly, the general LGBTI+ population” (p. 13) [23]. In response, the Strategy identified three overarching goals, with the third of these prioritizing the development of the research and data environment with a specific objective to “develop research into the factors that support positive mental health for LGBTI+ young people” [23] (p.31). 

### 1.2. Creating Supportive Environments

The importance of supportive social environments has been identified as essential in empowering individuals [2,3]. The Irish Department of Justice is currently developing an LGBT inclusion strategy to: “target discrimination, promote inclusion, and improve quality of life and wellbeing for LGBTI people” [24]. While such strategies are welcome, it has been acknowledged that homonegativity and transphobia do not disappear as a result legislative and policy changes [25]. Thus, broader supportive societal environments are recognized as key factors in promoting LGBT wellbeing. Such approaches have been embraced with the development of LGBT inclusion strategies by mainstream sporting bodies, for example [26]. Yet Abichahine and Veenstra found that while lesbian or bisexual women are more likely to involved in sport, the trend is reversed for gay and bisexual men [27]. Denison and Kitchen highlighted the gender-normative and homophobic experiences of LGBT-identified people within mainstream sport [28]. This may, in part, explain the reluctance to engage in sport within LGBT communities, particularly by gay, bisexual and transgender men. Ceatha has suggested that LGBT community involvement in interest sharing provides opportunities to challenge stereotypes, for example, regarding gay men and their interests [29]. Further, Browne, Bakshi and Lim suggest that LGBT community spaces, both formal and informal, may provide a sense of safety, beyond heteronormative and gendered assumptions [25].

Perhaps such experiences account for the global proliferation of groups, established by and for LGBT communities, over the past 45 years [30]. These groups include: Frontrunners, established in 1974, with over 100 LGBT athletics clubs worldwide [31]; *Various Voices,* a biannual LGBT choir festival [32]; the *Bingham Cup*, an annual international rugby competition [28]; a plethora of LGBT film festivals [33]; an International Theatre Festival [34]; and the *Gay Games* [35]. In applying recognition theory to these contexts, it is possible to understand how such social solidarity contributes to the creation of supportive environments established by and maintained for LGBT communities [2,8,9,10].

### 1.3. Strengthening Community Action

The literature tends to highlight the potential for social connectedness through LGBT communities to impact positively on LGBT identity and wellbeing [36,37,38,39]. Formby explored LGBT people’s understanding and experience of ‘community’ and the impact on self-reported wellbeing [36]. She found that most participants expressed a sense of belonging through an LGBT identity with ‘like-minded’ others, with others identifying collective LGBT identity as formed through shared experiences of discrimination [36]. Further, Detrie and Lease explored a sense of connection within LGBT communities where collective self-esteem tended to contribute to participants’ psychological wellbeing [37]. Similarly, DiFulvio found an association between positive self-identification, social support within LGBT communities and greater social and psychological wellbeing [38]. An alternative finding by Kertzner et al. highlighted the potential negative impact of LGBT social and community connections [39]. This, perhaps, may explain Formby’s finding that some participants resisted the commonly used term ‘LGBT community’ as a singular readymade entity which assumes an inevitable sense of belonging [36].

While not all people who identify as LGBT have a social connection to a singular ‘LGBT community’, Formby suggests a more appropriate usage may be reference to ‘LGBT communities’, which emphasizes the diversity ‘within and between’ those who identify as LGBT [36]. 

### 1.4. Developing Personal Skills

Building individual skills in promoting social, mental and physical health, it has been argued, involves sharing the requisite knowledge to empower, foster mastery and promote self-esteem. The *Ottawa Charter* emphasizes the need to strengthen the person within the context of a changed environment [2], recognising the structural constraints on LGBT agency. Wilkinson and Marmot concur, noting in the *Social Determinants of Health* that exclusion has a social meaning and impacts negatively on health and wellbeing [16]. This is consistent with Meyer’s concept of minority stress [4]. 

Androite highlighted DeVries’ emphasis on the importance of LGBT social connections in attenuating the impact of minority stress [40]. Others have argued that community connectedness may ameliorate the effects of stigmatization, prejudice and discrimination [37,38,39,40,41], a position which accords with Honneth’s ideas [8,9,10]. The concept of interpersonal recognition has the potential to develop individual resilience and promote self-confidence [9]. In addition, Heard, Lake and McCluskey suggested that the connection with others and their positive response through affirmation or encouragement, increases creativity and vitality with a subsequent increase in the sense of wellbeing [42]. Spencer and Patrick identify the importance of the concept of mastery and social support, which they suggest could account for differences in self-reported LGBT psychological wellbeing [43]. A similar conclusion was drawn by Ceatha in a project that found that involvement in LGBT community groups facilitated ‘mastering wellness’ through a broad understanding of wellbeing [29].

### 1.5. Reorienting Health Services

The *Ottawa Charter* provides for a holistic model of service provision premised on three health promotion strategies: advocacy for heath; enabling all to achieve their full health potential; and mediating between different interests [2]. The Institute of Medicine emphasized the challenges in addressing stigma, distinguishing between structural stigma and personally enacted stigma [44]. Formby noted the potential risk of heteronormative institutions and practices to impact negatively on LGBT identity and wellbeing [36]. Further, Haas et al. conceded that the current research focus limits the capacity to monitor and assess the full impact of policies and practices seeking to promote improved LGBT mental health and wellbeing [5]. This stigma, at both the structural and personal levels, may extend to a limited recognition of the cultural and social capital embedded within LGBT networks [45]. This is somewhat surprising given the inverse relationship between mental ill-health and social capital [46]. However, a commissioned report carried out in Ireland claimed that “LGBT people have less access to ‘social capital’” [45]. This failure by health services to recognize the individual and collective contribution of LGBT communities may inadvertently create barriers to healthcare access [47]. This suggests that in order to reorient health services, policy development and implementation of practice should be informed by research recognising the contribution of LGBT communities [3,5].

Given the exponential growth of groups, set up by, and for, LGBT communities [30], the lack of research into LGBT social connectedness through involvement in interest groups is surprising. The sparse literature on mental health promotion highlights the importance of understanding strengths-based community approaches that promote LGBT wellbeing. This study is aligned with emerging research trends calling for an exploration of protective factors that promote LGBT wellbeing [3,5,11,12,13,14,15,17,23].

## 2. Materials and Methods

Qualitative methodologies provide a nuanced picture of the meanings, understandings, and experiences of a social group [48]. These factors may remain hidden if quantitative approaches are used [49]. An exploratory qualitative approach sought to illuminate an under-researched topic with a view to “generating results and theories that are understandable and experientially credible both to the people being studied and to others” [49]. The research process was informed by an “iterative design” [50] from the framing of research questions through to describing, analyzing and interpreting the rich data.

### 2.1. Research Aims and Objectives

The research aim was to explore the social meaning and significance of LGBT social and community through their sporting, creative or social pursuits. Giving voice to LGBT people’s perspectives and priorities is of paramount importance, with the recognition of mental health inequalities for people from minority sexual orientation and gender identity populations [3,4,5]. The study prioritized the involvement of LGBT stakeholders and communities in recognition that some LGBT voices may be under-researched [51]. Drawing from Baker and Beagan’s emphasis on “learning *with*” LGBT communities [52], the research sought to problematise the assumption that LGBT communities are a universally vulnerable group [11,12]. This approach strengthened the research process, from design to dissemination, enhancing the quality of the study [51]. It also ensured diverse representation across age, sexual orientation, gender and gender identity, nationality and sporting, creative and social interests, informed by Rubin and Rubin’s concept of a “conversational partner” [50].

### 2.2. Access, Recruitment and Sampling

A range of LGBT stakeholders, known to the first author, identified potential participants and provided introductions, thus validating researcher credibility and facilitating access to participants who may be considered potentially ‘under-researched’. A sampling strategy, comprising a mix of purposive and targeted sampling techniques [50,51], was designed to identify appropriate places and contexts for the recruitment of study participants. LGBT people, aged 18 years or older, who were living in Dublin and were involved in sporting, creative or social groups were recruited. LGBT communities and participants were enthusiastic about the study. Due to the level of interest expressed by LGBT community groups and to ensure inclusion, 11 people participated in 10 interviews, with one joint interview. The sample sought to include a broad age range of LGBT people, with four people in their 20s, three in their 30s, two in their 40s, and two in their 50s. Although all participants resided in Dublin at the time of interview, a number were from rural localities and two were born outside Ireland.

### 2.3. Data Collection

The qualitative interview was the core data collection method informed by Kvale and Brinkmann’s description of an *InterView*; an inter-change of views resulting in the co-construction of knowledge [48]. Rubin and Rubin identify four core characteristics of ‘responsive interviewing’: 1. choosing interviewees who are knowledgeable about the research problem; 2. listening carefully to what they tell you; 3. asking additional questions until their answers are fully understood; and 4. seeking depth, detail and richness derived from interviewees’ first-hand experiences [50]. While the interview was largely unstructured, it focused on four broad areas: background, joining groups, wellbeing, and interests. A pilot interview was conducted to ensure shared meanings, clarify understanding of questions and identify any potential gaps in the interview schedule. A reflexive approach, where the researcher is aware of, and openly acknowledged, their role in the study and how they may affect the process, was adopted during the data collection and analysis processes [49].

### 2.4. Data Management and Analysis

Interviews were transcribed verbatim, with all identifying information removed from the transcripts. An open coding technique reduced the data into concepts and categories that were identified on an iterative basis [53]. The identification of themes woven through the interviews and the technique of linking categories aided the identification differences and similarities across participants’ accounts. The analysis sought to recognize diversity by accounting for negative instances and incorporating data that contradicted emergent or dominant ideas. The coding categories permitted the presentation of individual stories to give ‘voice’ to both typical and unique experiences [52]. Due to the exploratory nature of the research, the study did not seek to achieve ‘saturation’ [49].

### 2.5. Ethical Considerations

According to Maxwell, the four key ethical issues that require consideration in the conduct of research are informed consent; privacy; harm and future research [49]. The voluntary, informed consent of individuals was obtained prior to their participation in the study. Stakeholders disseminated information sheets outlining the purpose of the research and what participation involved. All participants were given between one and three weeks to consider whether they wished to participate, and it was explained that they could withdraw from the study without explanation or negative consequence at any time. Assurances of privacy and confidentiality were also provided. To ensure the anonymity of participants, pseudonyms were used and all potentially identifying information (names of places, people and so on) removed from the transcripts. In order to address potential harm, member checking was achieved through dissemination of the study findings and themes to participants and their feedback sought. Due to the ongoing interest in the study, LGBT stakeholder engagement facilitated dissemination of the research with participants, stakeholders and LGBT communities. Consent was sought throughout this process and specifically in relation to presentations sharing the research within LGBT sporting, creative and social groups. The involvement of LGBT communities from design to dissemination suggested directions for future research. Ethical approval was granted by a Research Ethical Approval Committee in Trinity College Dublin (REAC ref: 521).

## 3. Findings

### 3.1. The Sample

The eleven respondents included five lesbian women, four gay men, one bisexual person and one transgender person who were involved in 15 LGBT groups and 16 non-LGBT groups. Participants’ interests included physical activities (athletics, hiking, roller derby, rugby), creative pursuits (theatre, art, choir, creative writing) and social groups (dining, online MeetUp for socialising). The youngest participant was 22 years and the oldest was 56. The majority (*n* = 10) were fully open about their gender identity and sexual orientation, with all describing their annual participation in the Dublin Pride Parade, possibly reflecting their connectedness to LGBT communities. Most participants were in a relationship at the time of interview (*n* = 7), including civil partnerships and marriages; three of the study’s participants had children. Nine participants were Irish, while one was born in a European country and one in North America. All lived in the greater Dublin area and a majority (*n* = 10) had completed university education. A large number pursued their primary interest through LGBT groups (*n* = 7), with group membership ranging from between two and 14 years. Others have overlapping interests within and outside LGBT communities (*n* = 4). These four participants have long-standing involvement in their interests (5–28 years), with their involvement in LGBT groups being more recent, but extensive nonetheless (1–11 years). This article will focus on experiences within LGBT sporting, creative and social community groups.

Three key themes—‘connecting’, ‘mastering wellness’ and ‘making a difference’—were identified following a detailed analysis of the interview data. The following sections examine the theme of ‘connecting’, highlighting several forms of connecting, including through shared interests or skills; connecting as LGBT; connecting socially; and connecting with LGBT communities. 

#### 3.1.1. Connecting through Shared Interests

Interests played a pivotal role in the lives of all participants, with activities referred to as “central to nearly everything”, a “huge part of my life” and an “integral part of my day and my life as it “holds the key for everything”. Enjoyment of these activities was enhanced through these social connections. For some, group membership was “primarily about the people” who are “my type of people”. As one participant explained:
I think it’s important…as a part of a sense of self…to be a fully-rounded person…as part of your identity…you need to have something more in common than you’re just LGBT (MeetUp, 27)


This appears to disrupt Honneth’s idea of ‘identity politics’ and the “claim for recognition of one’s own culture” [10]. As such, LGBT community groups may create a space where sexual and gender minority identities are assumed, but which may not be defining, even if recognized by others. However, all respondents (*n* = 11) were involved in shared interests with others within LGBT community groups rather than as solo activities. Participants (*n* = 8) expressed a sense of belonging with like-minded others:
It’s the place to meet like-minded people…it was our love of food and wine-matching that was the connection…it connects me to the wider community…I connected it with being gay (dining, 56)
I’m part of this community and feel included and involved in it…because there’s lots of different types of like-minded people there…It’s a thing that the community really celebrates diversity in that sense…I just feel that I am accepted—it’s wonderful, it’s so validating (roller derby, 27)
But the main connection for everybody is the love of being outdoors on mountains or hills…Sharing my enjoyment of the activity…there’s an on-par thing now—a shared thing together…A coterie of like-minded people can go and do more adventurous things…and there’s a thrill in that (hiking, 53)


Shared interests appeared to play an important role in creating social networks and community connections. This concurs with Formby’s findings which highlighted a sense of belonging to LGBT communities with like-minded others [36] and is reminiscent of Honneth’s idea of the importance of interpersonal recognition [8,9,10]. It appears from these narratives that this recognition was reciprocated, offering a sense of connection, through the shared interests [9]. This has important implications for policy and practice in recognising the importance of such connections. While there is no singular way in which the connections with LGBT community groups are formed, albeit through food or hills, theses narratives suggest that feeling included and ‘part of something’ was critical.

#### 3.1.2. Connecting through Skills

The organic, yet exponential, growth of LGBT community groups in Ireland offer a wide range of activities and pursuits, perhaps reflecting the breadth of diversity ‘within and between’ LGBT communities [36]. Such groups appear to support the acquisition or development of personal skills [9,42,43]. Study respondents spontaneously mentioned the importance of having peers with a shared interest in understanding the activity and the challenges associated with the acquisition of new skills:
It evolved into a support network where we help each other by giving critical feedback on each other’s work. So, it’s evolved…and now it’s become really quite a solid unit that’s a support network for me (art, 23)
The people in the creative writing groups have a very good understanding of what it is I’m trying to do…It’s what I can achieve, what I want to achieve…It is honouring what somebody does well (creative writing, 44)
Just makes me happy…Yeah, quite relaxing, calming…and, from the choir point of view, fulfilling as well…just being able to achieve something, even small, at the end of a rehearsal (choir, 32)


In this way, the social networks formed through shared interests appeared to provide encouragement and motivation [42]. Participants frequently mentioned group members who recognized the effort involved in developing competence or mastery of the activity as providing affirmation:
The dynamic in the group is very supportive…If they see improvements, they say: ‘That’s a good one today.’ I’d be the same, I’d say: ‘I can see you are really coming on: the speedwork is really helping’ (athletics, 46)
Every time I tried something new I was getting better; seeing the recognition on people’s faces when they would see me in the rink and me improving as well… (roller derby, 27)


These narratives point to a sense of the importance of peer validation of achievements and the support provided in overcoming challenges in acquiring new skills [42]. Honneth also highlights the importance of mutual interpersonal recognition with others who share common interests, which can contribute to self-confidence [9]. In this way, the social networks explored in the study, appeared to provide an important form of recognition, described by Honneth as competence and community contribution [9]. This also reflects Spencer and Patrick’s notion of mastery and social support [43] and Ceatha’s concept of ‘mastering wellness’ [29].

For all of the participants (*n* = 11), the importance attributed by them to their shared interests was evident in terms of levels of engagement. This included attending training and workshops, personal investment, in time, equipment and travel; the level of support, both to others and the group; and the extent to which their interests have developed over time. Further, the recognition of the individual contribution of members within these communities appears to positively contribute to self-confidence for the individual and self-esteem for the collective [8,9,10]. While this has important policy and practice implications, it appears that outside of LGBT communities, such contributions have been largely unrecognized, particularly in the health policy arena.

#### 3.1.3. Connecting through LGBT Identity

It was evident from the interviews that, for a number of participants (*n* = 9), having shared interests facilitated entry to a group, established by, and for, LGBT communities. Respondents conveyed a nuanced understanding of the interconnection between their interests and their LGBT identity. Many, for example, identified their sexual orientation or gender identity as an incentive for joining a group established by, and maintained for, LGBT communities:
I found myself in a situation where I had no gay friends and it was a connection that I missed…I figured the only way to really make friends who are also LGBT was to join some sort of a club that had LGBT participants (MeetUp, 27)


In this way, it appears that it is the interlinking of the interest and LGBT identity that creates the connection; there is a need for both, one without the other would be insufficient. However, this is seemingly contested by those who feel that they do not necessarily connect to LGBT people or need LGBT-specific communities. This may accord Formby’s findings that challenged the assumption of the inevitability of a connection to a readymade ‘community’ [36]. An example of this is outlined below:
I think I’m defined first and foremost by the fact that I am really passionate about acting and really passionate about theatre and performance. That’s my main thing (theatre, 22)


Nonetheless, the presence and visibility of LGBT or gay people in ways that are normalized cannot be underestimated [10]. As such, in spaces where people feel a sense of connection they may not seek out ‘like-minded’ people around sexual orientation and gender identity, because they are already part of a world where they feel included and supported. While their interest is predominant, this participant expanded on this:
And plus is the fact that there’s always lots of actors who are gay, a lot of theatre makers who are gay; so it’s like I fit right in (theatre, 22)


In this way, it appears that while participants’ shared interests are to the forefront, these shared interests also provide opportunities for connection with LGBT communities. While there was evidence of “collective self-understanding” [10], other perspectives emerged:
First I said to myself ‘Ok, there is a gay rugby team, I’m not going to play for a gay rugby team just because I’m gay’ (rugby, 31)


However, this participant subsequently became involved through a tournament:
For the first time in my life I was socialising with all gay people…something just, I don’t know if it clicked, or if it felt comfortable…and I thought ‘Actually, this is pretty cool: I feel like I belong here, I feel at ease here, I feel comfortable’ (rugby, 31)


While the LGBT group provided a sense of belonging, further elaboration was provided:
I’d describe myself as…interested in sports…yeah, I’m gay…but I wouldn’t want it to be, and I don’t think that it should be, one of the first words that I’d choose to describe myself (rugby, 31)


This narrative suggests that while there was no ‘need’ for a gay team, as sexual orientation is only one form of identity enactment. However, connection with a gay team, through a shared interest in rugby, became a space of connection that moved beyond sexuality. As such, it was both the shared sporting interest and the inclusion through sexual orientation and gender identity that were critical factors. In this way, it appears that self-identification through shared interests provides an opportunity for the salience of these identities to be contextually determined. 

While Honneth suggests that the recognition of “rightful identity” [10] provides an impetus for emancipation, it appears that LGBT social networks provide spaces that extend beyond identification solely on the basis of sexual orientation or gender identity. Thus, recognition of the multiplicity of identities is enabled, with the sporting, creative or social interest of increased salience within the group. This finding suggests that it is the shared interests that encompass and expand the concept of a singular LGBT ‘community’ [36].

#### 3.1.4. Connecting Socially

Although Honneth specifically mentioned ‘identity politics’ in relation to LGBT communities, respondents’ accounts strongly suggest that they had a more nuanced understanding of the interconnection between their interests and their LGBT identity [10]. In particular, their social networks offered important forms of interpersonal recognition and a sense of belonging [9,16]. Such groups provided alternatives to the LGBT pub–club scene and, for some, this community connection was an incentive to join the group:
How do you get to know people?—You share an interest, join a sports team, join a choir… a big group of friends and family, but big and gay and in Dublin. [It] was something you could do that was gay and wasn’t just going to a bar... (choir, 36)


The importance of these groups in facilitating connection to LGBT communities at the time of ‘coming out’ was mentioned by three participants. The age of ‘coming out’ ranged from 16 to 44 years, with an average age of 23 years for the sample. Some participants were very clear about their age on ‘coming out’ while others described a process:
I loved it, I was going every two weeks, never missed it…it’s not a vehicle for ‘coming out’ and, in a sense, I did use it for ‘coming out’… I was very, very nervous going to my first meal and I’ve met a lot of people who have been very nervous…It has become where I’ve met my best friends and quite a lot of acquaintances (dining, 56)
When I found out there was a gay choir, it seemed obvious thing. People were so welcoming and really, really lovely. It was my first experience of lots of gay people who weren’t just the same age as me…I was immersed in it quite quickly—it was everything that I wanted it to be—a great way into the community…it wasn’t too much outside my comfort zone (choir, 32)


While none of the sporting, creative or social groups was depicted by participants as a support group when ‘coming out’, it may be that sharing interests acted as a conduit for those exploring and questioning their LGBT identity. Further, such supportive environments [2,3] may provide interpersonal recognition through LGBT connection at critical time points, including throughout the ‘coming out’ process [10]. 

For others, the social element within the LGBT community groups established by, and for, LGBT communities, provided the motivation to join:
One of the great things about it is…they meet for coffee and buns afterward and it is a lovely social occasion…even if you’re feeling crap you will plan your weekend around [the] Saturday morning run (athletics, 46)


It appears that the availability of the broad range of interest groups [30] facilitates the recognition of diversity within and between LGBT people [36]. This emphasises the importance of recognition of diversity across LGBT communities as a key aspect of social connectedness. LGBT sporting, creative and social groups seem to be affirmed by members for their promotion of the diversity of LGBT interests. The plethora of LGBT community groups established by, and for, LGBT communities attests to this [26,27,28,29,30,31,32,33,34,35].

#### 3.1.5. Connecting with LGBT Communities

Most participants felt that they were part of a wider network of LGBT communities, characterized as a “very welcoming and open club” which was “safe and accepting” and fostered feelings of being “included and involved.” The importance of safe spaces that were accepting is consistent with Browne, Bakshi and Lim [25]. In the context of perceived societal progress, there was recognition of the potential for present and future policies could impact positively on LGBT wellbeing in Ireland [8,9,10], particularly given the rapid sense of legislative attitudinal shifts that had occurred during the past decade. This echoes Honneth’s appeal for universal legal rights, empowering communities and supporting social integrity [8,9,10]. Equally, there was an awareness of the limitations of legislative changes [25]. As such, the importance of the empowerment of LGBT communities extended to challenging stereotypes (*n* = 6) and creating visibility (*n* = 6) through involvement in LGBT community groups:
Some people don’t know any gay people, or don’t know they know any gay people or it’s this ‘other’ thing and they see it as the stereotypes…We’re a choir who mostly happen to be gay (choir, 32)
…and a choir singing—especially a relatively traditional choir—is very non-threatening and quite accessible and quite different (choir, 36)


The discussion above, mentioning that the choir ‘mostly happen to be gay’ and ‘non-threatening’, suggests that others’ perceptions of LGBT people and social connectedness beyond LGBT communities is also important. However, while visibility matters, it appears to be premised on an expectation that a visible ‘non-threatening’ LGBT presence will be greeted positively and not result in heteronormative or gendered responses. This perspective was shared by a number of respondents who mentioned the importance of creating visibility (*n* = 6) to counter dominant representations of LGBT lives. Additionally, the importance of a visible presence focused on creating connections within and between LGBT communities [35]:
It is good to show up and have a presence, a visibility for LGBT people…If you see somebody representing the lesbian and gay community running at the same event you’re running at, that has to allow you to have some sense of belonging or affinity or potential to be…not so alienated (athletics, 46)
A lot of my friends have said to me that they felt like I did pave the way; that me ‘coming out’ gave them the courage to afterwards…I think young people now consider to be homophobic…to be just medieval. My generation doesn’t really care anymore what you are and that’s pretty good (art, 23)


This relies on specific understandings of a ‘new’ Ireland with a younger generation as inherently more open and accepting and who ‘doesn’t really care’ about others’ sexual orientation or gender identity. However, reflecting the contested and contradictory nature of these debates, other participants were less certain that societal progress was reflected in LGBT people’s lived experience:
There’s still a lazy portrayal of gays in the media…where being gay is everything about them or just with their mincing characteristics or they speak in one way and they’re flamboyantly gay, which…does exist…But…anyone, regardless of their temperament or how they act, can be gay (theatre, 22)


There were recurring references to homonegativity and the prevalence of the “presumption of heterosexuality”:
It’s the same, say for any say teenage boy growing up, when the person asks: ‘got a girlfriend?’ they never ask…‘or a boyfriend?’ When people assume I’m straight I nearly feel an obligation…to tell them I’m gay…It’s the way of life to be straight (rugby, 31)


Perhaps, due to such experiences of homonegativity, heteronormativity and gendered assumptions, some respondents felt it was important for their wellbeing to be involved with an LGBT sporting, creative or social group [26,27,28,29,30,31,32,33,34,35]. This is consistent with Browne, Bakshi and Lim’s suggestion of such groups providing safe spaces [25]. It may also be that LGBT groups serve an important purpose in providing a sense of belonging [16], promoting wellbeing [3,5,17] and ameliorating the effects of minority stress [38,39,40]. While it has been suggested that this contributes to the development of personal resilience [9,37,39], this too was questioned by one participant:
I think to make resilience the focus of the solution is to shift the responsibility from the perpetrator to the person who’s been affected by the behaviour…I’m very resilient. I shouldn’t have to be resilient all the time. My life shouldn’t be a battle (creative writing, 44)


Major events, such as ‘coming out’, forming or ending relationships, parenthood and loss through bereavement, were all cited as examples that participants felt created pressure in negotiating assumptions and stereotypes. It also appeared to provide an impetus for calls for recognition [10]. A considerable number of participants (*n* = 7) specifically mentioned the contribution of LGBT communities to these societal changes:
Those challenges and activism have actually led to the changes that we have now…The challenge is trying to change society to be more just and more diverse…whether that’s racially…or sexuality or gender—a thriving society is one that is diverse economically and creatively and socially generally (hiking, 53)


The potential contribution of LGBT community groups in creating formal and informal safe spaces [25] and establishing and maintaining groups that potentially ameliorate the effects of minority stress [39,40,41] may, in part, explain the longevity and continued demand for such groups [26,27,28,29,30,31,32,33,34,35]. This suggests that LGBT communities are pivotal in creating social networks and a sense of connection [9,16,36]. As argued by Honneth, LGBT social networks appear to play an important role in the continued empowerment of LGBT communities, with a positive impact on individual self-confidence and collective self-respect [8,9,10].

## 4. Discussion

This in-depth exploration of how LGBT people describe the meaning and significance of their shared interests, hobbies and pursuits reveals there is a lot to “learn *with*” LGBT communities [52]. The findings presented suggest that groups, by and for LGBT communities, are pivotal in raising aspects of recognition, both in terms of interpersonal and community contribution as well as right-based activities [8,9,10]. However, the contribution of LGBT communities in promoting wellbeing appears to have been largely unrecognized. This will now be discussed in light of Honneth’s tripartite framework of interpersonal, community and rights-based recognition [8,9,10].

### 4.1. Interpersonal Recognition

The people in this study actively sought avenues of social networking that facilitated engagement with interests that were personally significant and represented positive contributions to their lives. In many cases, respondents appeared to benefit from experiences of mutual recognition from like-minded others [8,9,10] because it provided affirmation, validation [41,42] and a sense of belonging [16].

Involvement with an LGBT group also provided safe spaces [25] and, as such, were a mechanism for circumnavigating gendered and heteronormative assumptions [36]. Respondents were acutely aware that social exclusion and societal treatment that is less than equal has the potential to impact negatively on their wellbeing and that of wider LGBT communities [2,3,4,5,6,16]. In this way, LGBT social and community connectedness may provide a buffer against the effects of minority stress [37,38,39,40,41]. These findings are consistent with Formby’s, where collective LGBT identity emerged as a bond that developed through a sense of belonging and, for some, through shared experiences of discrimination [36].

Participants’ understanding of their experiences, provide considerable insight into the unacknowledged, underlying influences on their wellbeing. The ability to exercise agency is the product of cultural meanings that are both constructed and constrained [54]. While respondents were adept at recognizing constraints on their agency, this recognition, in itself, may lead to a perception of choice. This underscores the emphasis placed throughout the *Ottawa Charter* on changing the environment alongside strengthening the person [2].

### 4.2. Community Recognition

There was a sense that respondents had often exercised agency in relation to self-definition and identity enactment through interest sharing. These provided opportunities for social and community connectedness [37,38,41]. Perhaps, through the creation of spaces where one’s LGBT identity is assumed, other forms of identity enactment, in such sporting, creative and social interests, are enhanced. In this way, LGBT groups, by and for LGBT communities, appear to both encompass and extend beyond what Honneth termed ‘identity politics’ [10]. The formation of these groups around shared interests may have provided opportunities for side-stepping Honneth’s suggestion of “collective self-understanding” and “claims for recognition” solely on the basis of sexual orientation or gender identity. Rather, through group membership, participants appeared to move beyond the potential for a limited and limiting monostaticity of identity.

These narratives point to the wealth of cultural and social capital embedded within LGBT networks. Contrary to claims that LGBT communities have less access to social capital [45], it appears such groups play an important role in recognising diversity within and between LGBT communities [36]. It may be that, in these circumstances, LGBT communities in Ireland appear to be at the forefront of mental health promotion [2,3,16]. With strengthened community capacity, LGBT interest groups may provide an example of the second goal of *Connecting for Life* [3] since empowered communities that respond to the needs of members can provide a buffer against the effects of minority stress [3,4,5,37,38,39,40,41] and help to sustain these benefits over time [42,43].

Through community membership of such groups, participants sought recognition of LGBT human rights through creating visibility and challenging stereotyped assumptions [8,9,10]. Despite this important contribution by LGBT communities, when viewed through the lens of Honneth’s tripartite framework [9], there appears to be a lack of recognition by policy makers of the strengthened LGBT community action in creating these supportive environments, which enhances the development of personal skills [2]. 

### 4.3. Recognition of LGBT Human Rights

While policy is specifically designed to address identified problems, it is possible that polices, and, therefore, practice, remains trapped in negative connotations of the problem [55,56]. Thus, the broad research consensus of elevated mental health risk [3,4,5,6,46,47], as a consequence of stigma and discrimination [3,4,5], may have inadvertently led to a tendency to focus on resultant deficits. Thus, people who are LGBT may be perceived only as the recipients of healthcare rather than as contributors to their own wellbeing and that of others within LGBT communities. This lack of recognition at interpersonal and community levels, as outlined in Honneth’s tripartite framework [9], may inadvertently reinforce structural and personally enacted stigma within health and social care policy and practice [44].

The dearth of literature on mental health promotion within LGBT communities has been acknowledged nationally [3] and internationally [5]. However, in the absence of such research, the direction of policy and practice for health promotion, prevention and intervention with LGBT communities, may be solely informed by the current focus on identifying risk factors for mental health [3,5,6,46,47]. This underscores an anomaly whereby policy makers generally recognize the importance of community connectedness [3,4,5,6,46,47], yet fail to recognize such community connectedness within LGBT communities [6,46,47]. This paper highlights the critical importance of recognition of LGBT social networks and calls for shifts in policy and practice frameworks which recognize the strengths embedded within LGBT communities.

### 4.4. Study Limitations

This was a qualitative exploratory study with 11 participants and did not seek to achieve ‘saturation’. As such, the findings presented cannot be claimed to be generalizable to other LGBT populations. However, by gathering in-depth accounts, the research provides important insights into LGBT people’s understandings of their social and community connectedness and the positive role that LGBT communities could potentially play in health care and health promotions policies. This suggests that further research, particularly mixed-methods research, is clearly needed, in order to identify factors that promote LGBT wellbeing, consistent with current policy [3,23]. Studies of this kind should seek to ensure the inclusion of people from diverse socio-economic and cultural backgrounds, as well as young people, particularly in light of recent research suggesting substantial vulnerabilities among LGBT youth [47].

## 5. Conclusions

This in-depth exploration of how LGBT communities describe the meaning and significance of their shared interests, hobbies and pursuits suggests there is a lot to “learn *with*” LGBT communities [52]. However, within policy, the contribution of LGBT communities in promoting wellbeing appears to have been largely unrecognized. Despite the limitations of the study, the findings provide valuable, alternative views that may contradict policy norms. A health promotion approach, informed by the *Ottawa Charter* and underpinned by Honneth’s *Theory of Recognition*, makes it possible to envisage some shifts, from individual health toward service reorientation paradigms. Messages from the study reinforce the central role that LGBT communities can play in the promotion of mental health and social wellbeing, with important policy and practice implications. It would appear that while LGBT communities are in some ways affirmed by members for their role in promoting LGBT wellbeing through social support and mutual reciprocity, this has been largely unrecognized in the health and social policy fields in Ireland. This study exemplifies the social and cultural capital embedded within LGBT networks and the contributory role of interest groups, established by, and maintained for, LGBT communities, in enhancing social connectedness. It may be that future studies in this area can shed more light on these policy-making lacunae, especially given recent trends towards social prescribing to promote community involvement and address social isolation. Finally, the use of alternative theoretical frameworks to enhance debates in this area, in particular, notions of rights-based recognition, where issues of social justice and identify are fore fronted is necessary if health promotion in this area is to be effective.

## Figures and Tables

**Table 1 ijerph-16-03636-t001:** Honneth’s Theory of Recognition [9].

Forms of Recognition	Interpersonal Relations	Community Relations	Legal Relations
Mode	Intersubjective	Community contribution	Universal rights and inclusion
Potential	Security and resilience	Valuing strengths and competence	Empowerment
Impact	Self-confidence	Self-esteem	Self-respect
Community impact	Social networks	Social solidarity	Social integrity

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
