# Peer review of "The Power of Recognition: A Qualitative Study of Social Connectedness and Wellbeing through LGBT Sporting, Creative and Social Groups in Ireland"

_ijerph, 2019, doi:10.3390/ijerph16193636_

Round 1

Reviewer 1 Report

Ceatha and colleagues in their report investigated the feelings of LGBT towards the society and their preferences. The study involves the limited subjects. The data is derived using interview methods. The study is well written and could be an important contribution in increasing social awareness towards the choices made by LGBT. The study is well within the scope of the journal. Despite of the limited sets of subjects, the authors should have add a paragraph stating how much information can shape policy maker’s decision towards LGBT’s right.

Authors should elaborate the discussion part. Why only limited number of subjects were involved (funding issue? social shyness?

How such information can help to reduce the STDs in the society?

What are the ways to decipher the information to larger audience other than publishing in a scientific journal?

Author Response

Thank you for your review and encouraging comments. We particularly appreciate your feedback of how the article can be improved, especially by ensuring that the conclusions are supported by the results, particularly in shaping policy-makers decision-making. To address your points we have expanded the article so that the important principles of the Ottawa Charter are underpinned by Honneth’s Theory of Recognition. This has focused the article more strongly within the policy context so that the analysis and discussion emphasise the need for policy and practice frameworks to recognise the strengths embedded within LGBT community networks.

We note your queries around the limited number of participants. It is not unusual for qualitative studies to have small numbers of participants. This is the case, particularly for exploratory studies, that do not seek to achieve ‘saturation’ (that is to conduct interviews until further data collection is unnecessary.) We have included additional information to explain the recruitment process and how sampling took place to ensure a broad representation (not in the quantitative sense, but in terms of age, gender and gender identity, sexual orientation, nationality and place of origin and across sporting, creative and social interests).

While there is broad concern around increasing rates of STIs, it is beyond the scope of this article to address this.

We welcome your question on how to disseminate the information to a wider audience, outside of traditional journal publishing routes. This article has benefited from emphasising the application to health policy. We have included further information on the concept of ‘learning with’ LGBT communities, as this enhanced the study. It also meant that the findings continue to be shared within LGBT communities through presentations. We can include more information about sharing the research with wider audiences through presentations and contributions to health and social care practitioners, health researchers and policy makers.

Thank you, again for your review and your encouraging comments that the study is well written and could be an important contribution and is well within the scope of the journal. We think the revised manuscript has been strengthened by addressing your comments and suggestions.

Reviewer 2 Report

While I enjoyed reading this paper and I think has merit, and   should be published, I wished the authors would have given a bit more information in the background, especially on the topics of Building Health Promoting Policy and Developing Personal Skills.

Author Response

Thank you for your review and encouraging comments. We particularly appreciate your feedback of how the article can be improved, especially by ensuring that the introduction provides sufficient background, especially on the topics of ‘Building Health Promoting Policy’ and ‘Developing Personal Skills.’

To address your points we have expanded the article so that the important principles of the Ottawa Charter are underpinned by Honneth’s Theory of Recognition. This has focused the article more strongly within the policy context so that the introduction includes the background and policy context. This has also helped give structure to the analysis and discussion. The article emphasises the need for policy and practice frameworks to recognise the strengths embedded within LGBT communities.

In relation to personal skills, this section contextualizes this within the the Social Determinants of Health and with reference to Heard, Lake and McCluskey’s work re: interest sharing.

Thank you, again for your review and your encouraging comments that you enjoyed reading the paper, think it has merit and recommend that it should be published. We think the revised manuscript has been strengthened by addressing your comments and suggestions.

Reviewer 3 Report

This is a qualitative study of 11 LGBT people about their involvement in LGBT and non-LGBT social/sporting groups in relation to their sense of social connectedness and wellbeing.

I found the paper an interesting read, and as always, fascinating to hear individual stories and experiences from diverse backgrounds and cultural contexts. The study aims, design and results were coherent and spoke well to the topic of enquiry.

The findings about social connectedness and wellbeing is of course not new, and there have been some rich and extensive research in queer studies and history about the development of community and sexual politics particularly from the mid-20th century in LGBTI communities globally and locally. Thus for the study to be an original contribution to knowledge, I feel that the authors should take the analysis of the findings a little bit further, either in terms of theoretical development or in its practical implications for policy. Both of which were only briefly mentioned in the introduction and discussion, but not quite rendered enough depth for either aspect to be “hard hitting”.

For example, the authors suggest ‘social capital’ and ‘resilience’ as important aspects of LGBT social connectedness and wellbeing. However, they seem to come a little short in developing either of these aspects. The authors could perhaps elaborate a little more on the context and essence of ‘social capital’ in this study (perhaps with engagement with work of either Putnam or Bourdieu), which may take the study implications a bit further. The scholarship on ‘resilience’, particularly Michael Ungar’s work which tackles the interaction between ‘agency’ and ‘environment’, which the study authors also touch on, might be another suitable angle to underpin and better elaborate the significance of the study findings.

The practical implications in terms of policy and practice for LGBT health could be another angle to enrich the analysis. Perhaps a bit more discussion around the cultural and political context of the Irish/Dublin LGBT community, and the barriers and facilitators of community engagement could shed light on why and how some LGBT people are more able to take advantage of social engagement, while others may not; and what further support from policy makers and the healthcare community might further enhance social connectedness, having learnt from the ‘positive’ stories in this study.

There were some minor points of clarifications that the authors could also make:

The study participants were involved in both LGBT and non-LGBT groups – some explanation or results about how the experiences of being in these groups differed (or not) would be interesting The abstract reported the recruitment of 11 participants but only 10 interviews – was there a reason for this anomaly? This was not explained in the results section. Some proof reading on p. 9, line 329.

Author Response

Thank you for your review and encouraging comments. We particularly appreciate your feedback of how the article can be improved, especially by ensuring that the results are clearly presented and that the conclusions are supported by the results.

While we were heartened by your comments that the study aims, design and results were coherent and spoke well to the topic of enquiry, we welcome your suggestion that in order to be an original contribution to knowledge, the analysis should be enhanced either in terms of theoretical development or in its practical implications for policy.

Careful consideration has been given to your suggestion that Putnam, Bourdieu or Michael Ungar’s work could provide a useful theoretical framework. While all these approaches have merit, and are certainly theorists that would be beneficial in taking forward for future research, we also wanted to address your suggestions about the policy implications of the research. In order to do this we have expanded the article so that the principles of the Ottawa Charter are underpinned by Honneth’s Theory of Recognition. In the introduction we outline his tripartite framework highlighting three interlinking forms of recognition: interpersonal, community and rights-based recognition. It has enabled us to focus the article in the context of the need for recognition of the contribution of LGBT communities within policy and practice frameworks.

We have also taken on board your suggestion of including more references to the cultural and political context. Again, Honneth’s framework has provided a lens for this, specifically in relation to policy. This has provided a very useful basis for the analysis and discussion with specific reference to the implications of this research.

While we agree that explanations of the similarities and contrasts between involvement in LGBT/non-LGBT groups would be interesting, this is beyond the scope of this article. However, it offers suggestions for a future article.

Thank you for your query regarding the reporting of 11 participants and 10 interviews. This has been corrected in the abstract (including one couple) and the methods section (joint interview).

The error that was on p. 9, line 329 has been corrected.

As a result of responding to your suggestions, we believe the article has been strengthened. Thank you again for your review and your encouraging comments that you found the paper interesting to read and fascinating to hear the individual stories of the participants. We very much appreciate your comments and believe that addressing these has strengthened the article.  

Round 2

Reviewer 1 Report

I have gone through the authors rebuttal letter. I am satisfied with their responses. I congratulate authors for the good job. The manuscript may be accepted for the publication in the present form.

This manuscript is a resubmission of an earlier submission. The following is a list of the peer review reports and author responses from that submission.